# Interaction of Human Serum Albumin with Uremic Toxins: The Need of New Strategies Aiming at Uremic Toxins Removal

**DOI:** 10.3390/membranes12030261

**Published:** 2022-02-25

**Authors:** Fahimeh Zare, Adriana Janeca, Seyyed M. Jokar, Mónica Faria, Maria Clara Gonçalves

**Affiliations:** 1Departamento de Engenharia Química, Instituto Superior Técnico, Universidade de Lisboa, 1049-001 Lisboa, Portugal; fahimeh.zare@tecnico.ulisboa.pt; 2Centro de Química Estrutural (CQE), 1049-001 Lisboa, Portugal; 3Center of Physics and Engineering of Advanced Materials (CeFEMA), Laboratory for Physics of Materials and Emerging Technologies (LaPMET), Chemical Engineering Department, Instituto Superior Técnico, Universidade de Lisboa, 1049-001 Lisbon, Portugal; adriana.janeca@tecnico.ulisboa.pt (A.J.); monica.faria@tecnico.ulisboa.pt (M.F.); 4Department of Chemical, Petroleum and Gas Engineering, Shiraz University of Technology, Shiraz 71557-13876, Iran; s.m.jokar@gmail.com

**Keywords:** chronic kidney disease (CKD), protein-bound uremic toxins, OAT, OCT

## Abstract

Chronic kidney disease (CKD) is acknowledged worldwide to be a grave threat to public health, with the number of US end-stage kidney disease (ESKD) patients increasing steeply from 10,000 in 1973 to 703,243 in 2015. Protein-bound uremic toxins (PBUTs) are excreted by renal tubular secretion in healthy humans, but hardly removed by traditional haemodialysis (HD) in ESKD patients. The accumulation of these toxins is a major contributor to these sufferers’ morbidity and mortality. As a result, some improvements to dialytic removal have been proposed, each with their own upsides and drawbacks. Longer dialysis sessions and hemodiafiltration, though, have not performed especially well, while larger dialyzers, coupled with a higher dialysate flow, proved to have some efficiency in indoxyl sulfate (IS) clearance, but with reduced impact on patients’ quality of life. More efficient in removing PBUTs was fractionated plasma separation and adsorption, but the risk of occlusive thrombosis was worryingly high. A promising technique for the removal of PBUTs is binding competition, which holds great hopes for future HD. This short review starts by presenting the PBUTs chemistry with emphasis on the chemical interactions with the transport protein, human serum albumin (HSA). Recent membrane-based strategies targeting PBUTs removal are also presented, and their efficiency is discussed.

## 1. Introduction

Chronic kidney disease (CKD) can be recognized by its glomerular filtration rate (eGFG) below 60 mL/min/1.73 m^2^ or pathological albuminuria, and is characterized by progressive and irreversible nephron loss, reduced renal regenerative capacity, microvascular damage, metabolic changes, oxidative stress, and chronic inflammations, ultimately resulting in fibrosis and kidney failure (Figure 1). Myocardial infarction and stroke are common comorbidities.

CKD is a threat to public health around the world and is ever more frequently cited as a cause of morbidity and mortality and as an important risk factor for cardiovascular disease This recognition serves to illustrate very tidily the flaws in current therapeutic treatments. The burden of CKD has been studied in high-income countries, which are home to the world’s oldest populations (i.e., those aged 65 and above). Among CKD patients over 65 years, 30% develop stable disease, while the other 70% suffer from severe decline, with death, kidney failure, myocardial infarction, and stroke all increasing in frequency [2]. Moreover, 60% of people aged over 80 years developed CKD [3]; as the number of people aged 60 years and over is likely to double over the next 35 years, the frequency of CKD is expected to increase sharply [3]. Furthermore, by 2050, 80% of the world’s older people will be living in low- and middle-income countries, adding economic constraints to future CKD healthcare protocols. Sadly, younger CKD patients typically experience a progressive loss of kidney function.

Those persons suffering from CKD live with the constant threat of AKI, which is a major contributing factor in CKD progression [1,4]. The three most common causes of CKD are diabetes (accounting for 37% of all incidences), hypertension (30%) and chronic glomerulonephritis (12%), while the remaining causes can be attributed to cystic kidney diseases, interstitial nephritis and obstructive nephropathy. Drugs may also play a nephrotoxic role. Antibiotics, diuretics, contrast media, NSAIDs (nonsteroidal antiinflammatory drugs), ACEI/ARBs (angiotensin-converting enzyme inhibitors/angiotensin II receptor blockers), are just some of the examples. Diuretics were added recently to the list, accounting for 22.2% of all drug-induced AKI, ranking only after antibiotics [5].

Healthy kidneys play a vital role in the metabolism and processing of most drugs, as well as being essential in filtering toxins and waste products from the blood, controlling blood pressure and electrolyte functions, and controlling the functions of other bodily fluids. When renal function is impaired, a range of substances are retained in body, some of which are bonded to the transport protein human serum albumin (HSA). Those substances that remain in the body, which play a huge role in helping develop and manifest uremic syndrome, then become uremic toxins (UTs). Much about uremic syndromes remains unexplored, including their very nature and the relative importance of their components. As well as further research, more advanced removal systems than the ones that are available at present are urgently necessary.

## 2. Uremic Toxins

Developing new models related to pharmacokinetics, including cellular transport, albumin binding, and their effects on nonrenal drug clearance, will be key in designing new and effective treatment methods that will better combat the health problems brought about by UTs.

### Uremic Toxins Classification

Retention solutes are sorted by the European Uremic Toxin Work Group (EUTox) into three groups (illustrated in Table 1, Table 2 and Table 3), according to their molecular weight and serum protein-binding characteristics [6].

Table 1 summarizes the first group, which is characterized by low molecular weight (<500 D) and non-known protein binding (small free water-soluble compounds) [7,8,9,10,11,12,13,14,15]. Table 2 illustrates the second group, which is also characterized by low molecular weight (<500 D, leptin and retinol-binding protein being two exceptions, with medium molecular weight) but with known protein-binding (i.e., belonging to groups that are able to protein-bind [7,16,17,18,19,20,21]). Finally, Table 3 shows the third group, which is comprised of middle molecular weight compounds [7,22,23,24]. Together, 68 solutes have molecular weight below 500 D; among the 22 medium weight molecules, 12 (54.5%) have a molecular weight > 12,000 D [7,16].

UTs can also be classified according to their molecular ionic character in anionic UTs and cationic UTs (Figure 2).

## 3. Albumin: Structure and Physiological Functions

Transport human serum albumin (HAS) is a small globular protein (66.5 kDa) with an ellipsoidal shape (~140 × 40 Å) and an isoelectric point of 4.9, meaning that it is able to provide around 80% of the colloidal osmotic activity of normal plasma.

HAS is the primary protein present in human blood plasma, making up around 50% of human plasma proteins. Several other transport proteins exist in blood plasma, but albumin (despite its strong negative charge) appears to be capable of binding to a wide variety of compounds, whether they are positively or negatively charged (endogenous or exogenous) reversibly, with high affinity. Water, various cations (such as Ca^2+^, Na^+^ and K^+^), metal ions (Cu^2+^ and Ni^2+^), fatty acids, bilirubin, hormones (thyroxin, T4), vitamins (vitamin D), and pharmaceuticals (including barbiturates) are some of the examples [25]. Compound biologic activity, distribution, and clearance rate are limited by albumin-binding due to its effectiveness in reducing free concentration.

Structurally, HAS is formed by three homologous helicoidal domains (I, II, III), each made up of two subdomains with common structural motifs (sites I and II), arranged to form a heart-shaped molecule (Figure 3) [26]. Different isomeric forms do exist. The ability to fluctuate between isomeric HAS forms, in aqueous solution, could assist in adapting the albumin molecule to bind ligands with a diverse nature with high affinity. Undoubtedly serum albumin serves a role in the transportation of a large number of substances, but also serves an equally vital role in buffering or regulating the concentrations of various low molecular weight substances.

## 4. Organic Anionic UTs

Organic-anionic UTs have a major part to play with respect to the uremic syndrome. The deterioration of residual renal function in CKD [28,29,30] leads to comorbidities such as lesions in the veins [31,32,33], orthopaedic disorders [34,35], and severe nervous system conditions [33,34,35,36] (Table 4). Reduced immunity is a major cause of morbidity and oxidative stress in ESRD and is perhaps itself caused by uremic toxicity. Uremic cachexia is an underrecognized uremic syndrome.

Organic anionic UTs, such as indoxyl sulfate (IS), 3-carboxy-4-methyl-5-propyl-2-furanpropanoic acid (CMPF), p-cresyl sulfate (PCS), indoleacetic acid (IA), and hippuric acid (HA) (Figure 2) have low molecular weights, which allow their removal through HD. However, their strong bonding to HAS inhibits this clearance process.

Organic anionic transporters (OATs) mediate the cellular transport of UTs. When UTs accumulate intracellularly, they stimulate the production of proinflammatory cytokines spikes due to the production of reactive oxygen species (ROS), which themselves have a major role in CKD (Figure 4) [36].

A proinflammatory impact has also been shown to have been brought about by several cationic guanidino compounds [24].

### Organic Ionic Transporters

In the nephron, transporters, which are much involved in transporting substances such as inorganic ions, urea, phosphorus, steroid hormones, fatty acids, sugars, amino acids, peptides, and many clinically applied drugs (Figure 5a) [36], are diffused through the brush-border and basolateral membranes of renal tubular epithelial cells (Figure 5a) [36]. In humans, researchers have identified a superfamily of transporters, which can themselves be sorted into two groups, an adenosine triphosphate (ATP)-binding cassette series and a solute carrier series, which do not utilize energy derived from active ATP hydrolysis (Figure 5b) [37,38].

Vaziri et al. [39] illustrated the relationship between the UTs and dysbiotic gut microbiota. Koppe et al. [40] showed that probiotics expose normal intestinal microbiota, leading to reduced production of UTs [41].

The two transporter types that are of interest in this study, organic anion transporters (OATs) and organic cationic transporters (OCTs), can be classified by the organs and tissues from which they are expressed, or by their distinct functions. To date, six isoforms of OAT with different functions have been identified. Researchers have recently proved that the first type is expressed in a way that causes them to consist of three or four clusters tied by scaffold proteins immediately beneath the cell membrane [37,38]. Meanwhile, OATs in the kidney can be expressed in either the brush-border or basolateral membranes. The most characteristic function of OATs or OCTs is the transport of pharmaceutical drugs and their metabolites. The OAT transports drugs in the distal renal tubule to cells in such a way that it can provoke nephrotoxicity. OATs are also much involved in drug interactions, where more than one drug is excreted competitively, and are thought to play a role in the pathophysiology of uremia. Their involvement in genetic diseases has been clarified [36]. Also, they are thought to be involved in the progression of CKD and in the pathophysiology of uremia [36].

## 5. Pharmacokinetics of UTs

In this section, the different mechanisms involved in the transport of UTs are illustrated. Information shown in Figure 6 and Table 4 would be helpful in understanding the differences between mechanisms. IS, CMPF, PCS, HA, IA, are transported via OAT1 and OAT3 to various organs and tissues, while CTN and G are transported via OCT1 and OCT2 to various organs and tissues.

### 5.1. Indoxyl Sulfate (IS)

Bacteria such as E. coli produce tryptophanase, a digestive enzyme that brings about the production of indole from dietary tryptophan [28] in the digestive tract. When this indole is absorbed from the gut, it forms IS in the liver via hydroxylation by CYP2E1, thanks to sulfotransferase enzymes [42,43]. A small protein-bound molecule easily retained in uremia, IS is resistant and difficult to clear via traditional dialysis [28,44]. IS circulates in the blood stream and eventually causes nephrotoxicity (Figure 7) [35]. It is assumed in clinical practice that IS nephrotoxicity can be reduced by stopping the intestinal absorption of the indole. Researchers have also witnessed the expression of OATs in the capillary epithelium of the blood–brain barrier (BBB) and muscle cells.

Medical evidence supports the conclusion that IS is a UT that affects both the kidney and the cardiovascular system. Yamamoto et al. [42] demonstrated (in vitro) a vascular smooth muscle proliferation caused by IS, an outcome that can be prevented with inhibition of OAT3. Dou et al. [43] used a HUVEC (human umbilical vein endothelial cell line) to show that IS induces ROS in endothelial cells via a rise in NADPH oxidase (nicotinamide adenine dinucleotide phosphate oxidase) activity. In addition, IS alters the redox status of mesangial cells, causing production of both intracellular and extracellular ROS [44], which in turn is a cause of progressive renal failure.

The orally administrated charcoal sorbent AST-120 is already known to reduce IS levels. In a 5/6 nephrectomy rat model, Shimoishi et al. [45] showed that in vivo treatment with AST-120 brings down oxidized albumin levels and impedes the progression of CKD. Owada et al. [46] fed healthy and sub-totally nephrectomized rats with IS. Both rat populations (healthy and with CKD) showed lessened renal superoxide scavenging and worsened renal function with a high IS diet, suggesting that IS acts as a nephrotoxin system in the kidney. Taki et al. [47] presented data from 224 hemodialyzed patients, which showed raised IS levels and markers of cardiovascular disease, including calcium phosphate product, low levels of high density lipoprotein, and the advanced glycation end product (AGE) pentosidine [16,28].

#### OAT and IS

In a healthy kidney, elimination of IS from the plasma occurs by discharging IS via OAT. OAT1 and OAT3 move into the renal tubular cells, which can be found in the proximal basilar cell basement membrane (the basolateral membrane), and end up in the renal tubules located at the apical membrane of renal tubular cells (via OAT4) [48]. During renal failure crisis, UTs can travel via OATs to various organs and tissues, building up in both the kidney and the central nervous system, as well as in the placenta and muscles, potentially causing a wide variety of medical conditions (Figure 8).

Deguchi et al. [49] studied the pharmacokinetic and tissue distribution of IS in 5/6 nephrectomized rats (chronic renal failure rats, i.e., CRF rats) and evaluated the IS uptake by rat renal cortical slices, in vitro. The removal of IS from the blood plasma was less comprehensive in CRF rats than in sham-operated rats [t1/2β in the CRF rats (526 min)]. Elimination was more successful by a factor of 10 in the sham-operated rats, which excreted the majority of the intact IS via their urine. In renal cortical slice experiments, the IS uptake was a saturable process with a Km of 43.0 µM. Furthermore, sulfate conjugates, such as estrone sulfate (ES) (Ki = 25 µM) and dehydroepiandrosterone sulfate (DHEAS) (Ki = 16 µM), inhibited the IS uptake to a greater extent than p-amino hippuric acid (PAH) (Ki = 115 µM). PAH is a relatively selective substrate for rOat1 (rat Oat1) [50] while ES has a higher affinity for rOat3 (rat Oat3) [51,52]. This would seem to suggest that rOat1and rOat3 are important players in renal IS uptake. This was supported by indexing in vivo kidney uptake in rOat1- or rOat3-expressing systems [48,53,54]. In addition, rOat1 and rOat3 play roughly equal roles in contributing to renal IS uptake [54]. The human isoforms of rOat1 and rOat3, hOAT1 and hOAT3, are mainly expressed in the kidney and are found in the proximal tubules’ basolateral membrane.

Enomoto et al. [55] showed that hOAT4 plays some kind of role in the human proximal tubule apical efflux of IS. Cerebral concentrations of IS have been found to be 3.4 times than those in blood serum [56]. The source of this IS on the brain is not clear, but this limited distribution could be attributed to the brain-to-blood OAT of IS at the BBB. Dysfunctions, and the relevance of IS therein, have been evaluated by Oht Sukietal [29]. The brain-to-blood transport of IS at the BBB was investigated by the authors using the brain efflux index method, which was the role played by rOat3 in IS transport using Xenopus oocytes that expressed rOat3. The results showed that rOat3 mediates IS transport brain-to-blood, as well as being involved in the transport of neurotransmitter metabolites (such as homovanillic acid and 3methoxy-4-hydroxymandelic acid) and drugs (such as a cycrovir, cefazolin baclofen, 6-mercaptopurine, benzoic acid, and ketoprofen) (Figure 8) [6]. It follows that a stimying of rOat3-involved brain-to-blood transport occurs in uremia and leads to the prevalence of neurotransmitter metabolites and drugs in the brain (Figure 9) [6].

### 5.2. Carboxy-4-methyl-5-propyl-2-furanpropanoic Acid

3-Carboxy-4-methyl-5-propyl-2-furanpropanoic acid (CMPF), a furan dicarboxylic acid, was first conclusively identified in urine [57]. More recently, it was quantified in uremic plasma, where it accumulates in concentrations up to 60–370 µM [58]. CMPF’s pharmacokinetics and tissue distribution in normal rats were investigated [59], and the rate of contaminant elimination from plasma (after intravenous administration) was found to be:IS (t1/2β = 22 min) > A (t1/2 β = 65 min) > IA (t1/2 β = 122 min) > CMPF (t1/2 β = 356 min).

The renal and biliary clearance of unbound CMPF strongly suggests that the main elimination pathway is via renal excretion, through active tubular secretion. When we examine renal cortical slices, a mutual inhibition between CMPF and PAH is observed. In addition, Costigan and Lindup [30] reported that a CMPF plasma clearance decreased with PAH and probenecid coadministration, indicating a competition for the same transport system. The contribution of rOat1/hOAT1 and rOat3/ hOAT1 to the renal CMPF uptake was studied using rOat1- and rOat3-expressing LLC-PK1 cells or hOAT1- and hOAT3-expressing HEK293 cells in rat kidney slices [54]. The saturable uptake of CMPF by rOat1 and rOat3 was observed with Km values of 154 µM and 11 µM, respectively. The Km value for CMPF uptake by kidney slices (22 µM) was comparable with that of rOat3. In addition, CMPF uptake was preferentially inhibited by pravastatin and PCG (inhibitors of rOat3), which may indicate that the bulk of kidney CMPF uptake is OAT3. The in vivo transport mechanism responsible for CMPF renal uptake was evaluated by index methods [60]. The results show that rOat3 accounts for about 65% of the CMPF uptake in the in vivo physiologic conditions in rats. The brain-to-blood across BBB was also studied by the brain efflux index method. The saturable efflux transport of CMPF was inhibited by benzylpenicillin, taurocholate, or digoxin, which are substrates for rOat3 and rOatp2, respectively. Therefore, rOat3 and rOatp2 are involved in CMPF efflux across the BBB [6,61].

### 5.3. P-Cresol Sulfate

P-cresol sulfate (PCS) is a small molecule with its derivation in ingested phenylalanine and plant phenols. It is highly protein-bound and thus poorly cleared with HD [62]. Bammens et al. [63] first correlated high p-cresol levels and mortality in dialysis patients. They warned, however, that the p-cresol might not be the sole cause of toxicity, but rather a mere marker for the group of protein-bound retention solutes. They went on to find a correlation between higher p-cresol levels and cardiovascular events, thus indicating that p-cresol, or another protein-bound solute, may itself be a novel cardiovascular risk factor. A caveat should be added that it is the sulfated form of PCS the solute that is retained [62,64]. Schepers et al. [64] presented the first study on the biologic toxicity of PCS, supporting its role as a possible cardiovascular risk factor. In vitro, the oxidative burst activity of leukocytes rose with PCS, indicating a proinflammatory process that could be a high contributor to the occurrence of vascular disease in uremic patients [28].

Several works have indicated an association between higher free p-cresol serum concentration and cardiovascular disease (CVD) and mortality in haemodialyzed patients [27,63]. Some recent advances in analytical technology have indicated that most p-cresol (biosynthesized from dietary tyrosine and phenylalanine) goes through sulfate conjugation by a sulfotransferase enzyme while passing through the intestinal membrane. This process results in the production of a conjugate, PCS [65], which, in contrast to p-cresol, circulates in the blood [62,64,66]. HD removes PCS inefficiently because of its binding to albumin (90%) [63]. The same authors studied the PCS uptake using rat renal cortical slices and human proximal tubular cells (HK-2). The PCS uptake was saturated with mean K_m_ of 231.6 µM; the authors went on to look into PCS uptake, which told them that active transport is involved in the basolateral PCS uptake. They observed similar outcomes in HK-2 cells. This uptake was significantly suppressed by inhibitors of OATs, such as probenecid, PCG, PAH, and ES. Stymying was noticed when IS and CMPF were involved. The inhibitory effects of PCG and ES were larger than that of PAH. In contrast, only a minor inhibitory effect was observed for digoxin and tetraethylammonium, typical inhibitors of the organic anion transporting peptide (OATP) and the organic cation transporter (OCT), respectively. These data indicate that the OATs system plays an important role in basolateral PCS uptake in the kidney [6].

### 5.4. Hippuric Acid

Hippuric acid (HA) and indoleacetic acid (IA) are eliminated mainly by urinary excretion through active tubular secretion [59,60]. Studies using stable transfectants such as rOat1/hOAT1 or rOat3/hOAT3 cells prove that OAT1 mainly accounts for HA and IA renal uptake [54]. In addition, by kidney uptake index methods, the same authors demonstrated that rOat1 could be the primary mediator of renal HA and IA uptake, accounting, respectively, for ~75% and 90% of the transport (in in vivo physiological conditions) [60]. On the contrary, the renal clearance of endogenous HA appeared to be a useful indicator of changes in renal secretion that followed the reduced levels of rOat1 and rOat3 protein in 5/6 nephrectomized rats. Here, the renal clearance of unbound HA seemed to be closely correlated with the clearance of unbound PAH and not to the clearance of creatinine (CTN) [67]. HA in serum or cerebrospinal fluid is positively correlated with neurophysiological indices [68], suggesting that HA induces neurological symptoms, perhaps via inhibition of the OAT system at the BBB [29] or blood-cerebrospinal fluid barrier [69]. Similar to the efflux of IS, the efflux of HA and IA from the brain to the blood stream across the BBB appears to be mediated by rOat3 but not by rOatp2 [6,61].

### 5.5. Guanidino Compounds

An important role is played by several guanidino compounds (GCs). Vanholder et al. [7] classified GCs as free water-soluble low-molecular-weight solutes, rather than protein-bound solutes. The majority of GCs are difficult to remove by dialysis, in defiance of their weak protein-binding energy and large distribution volume [69]. At least four of them (including CTN, guanidine (G), guanidinosuccinic acid (GSA), and methylguanidine (MG)) can be found in high volumes in the cerebrospinal fluid of uremic patients [15,69,70]. It was demonstrated by Deyn et al. that the excitatory effects of uremic GCs on the CNS can be due to the activation of N-methyl-D-aspartate receptors, the inhibition of y-aminobutyric acid type A receptors by uremic GCs, and other depolarizing effects. It had been previously thought that G and CTN, which have within them a guanidine group, were moved around mainly by human organic cation transporter 2 (hOCT2) rather than by hOCT1 [71,72]. In this case, then, the excretion of GCs could be brought about through a deficiency of hOCT2 [73]. Toyohara et al. [73] reported that overexpression of human kidney-specific organic anion transporter OATP-R (SLCO4C1) in the rat kidney decreased the plasma level of GSA and another UTs, asymmetric dimethylarginine (ADMA), by metabolomic analysis using capillary electrophoresis-mass spectrometry. Here, it was found that the human SLCO4C1 promoter region has tandem xenobiotic responsive element (XRE) motifs, and various hepatic hydroxymethyl glutaryl-CoA reductase inhibitors (statins), which induce nuclear aryl hydrocarbon receptors, and upregulate SLCO4C1 transcription. Among the statins, pravastatin significantly increased the mRNA levels of rat SLCO4C1in the rat kidney in 5/6 nephrectomy (CRF rats). Here, the ADMA clearance rose noticeably in pravastatin-treated CRF rats without changing the CTN clearance, although the GSA clearance was not statistically significant. It seems clear from these data that, partially due to their length of service and high safety/tolerability profile, statins’ induction of SLCO4C1 into the kidneys of CKD patients may represent a new therapeutic tool that can assist in the excretion of uremic toxins and in reducing renal inflammation [6].

## 6. UTs Binding to Serum Albumin

The protein binding of endogenous compounds and exogenous drugs impairs, via competitive binding, leads to the accumulation of albumin-bound toxins, such as IS, IA, HA, and CMPF [74,75,76]. Watanabe and Hand did important work characterizing the binding of these UTs to human serum albumin (HSA), finding that the primary binding constant for each toxin could be expressed as follows:IS, Ka = 16.1 × 10^5^ (M^−1^); IA, Ka = 2.1 × 10^5^ (M^−1^); HA, Ka = 0.1 × 10^5^ (M^−1^); CMPF, Ka = 130.5 × 10^5^ (M^−1^)

It has been shown in competitive experiments and X-ray crystallographic analyses that while CMPF binds to site I, IS, IA, and HA bind to site II (these sites being the two principal ligand binding sites on HSA (Figure 2) [65,77]. Although the values of the binding constants listed above reveal that the interactions between the different UTs and HSA are relatively weak, it is important to note that the intensity of interaction and the stability of the resulting HSA-UT complex is strongly influenced by the surrounding microenvironment. Several studies on the binding mechanisms of HAS and UTs and pharmaceutical drugs have shown that the binding properties of HAS are strongly affected by factors such as temperature [78], hydrophobicity [79], pH [80], and ionic strength [81]. Furthermore, it has been shown that when studying the binding of a specific protein ligand, the interactive association of other UTs as well as pharmaceutical drugs that bind simultaneously to HSA can change the HSA binding properties and potentially modulate the final HSA-ligand binding mechanism [82].

As for CKD patients, reports have emerged of an impairment in the binding of bilirubin to HSA. Based on observations of a binding displacement and a molecular docking model, we can see that CMPF surely contributed to the defective binding of BR in uremia, since CMPF and BR share the binding localization for dicarboxylate molecules on HAS [83,84]. When it comes to PCS’s binding characteristics, which are not yet fully understood, some controversies have emerged. In vitro spiking experiments, using serum from haemodialyzed patients, were carried out by Meijers et al. [85], who found that when IS was added, the free concentration of PCS rose, and vice versa. This finding suggests that the same binding site on HSA is shared by both sulfate-conjugated UTs. It has been suggested in a recent clinical study that serum-free concentrations of sulfate-conjugated UTs (such as IS and PCS) might somehow be implicated in cardiovascular outcomes, CKD progression, and haemodialysis efficacy [86,87,88,89].

As mentioned, the environment can strongly affect the binding properties of HAS. Yu et al. [78] studied the interaction of HSA with IS in aqueous solutions with different ionic strength and at different temperatures. The results showed a decreasing binding affinity with an increasing temperature and higher ionic strength [78].

## 7. Effect of UTs on Nonrenal Drug Clearance

Drugs excreted renally are, unsurprisingly, only prescribed for patients with kidney problems in small doses. However, it may be surprising that even 25% of the medications that are eliminated by metabolism or nonrenal transport have an approximately twofold increase in the area under the curve (AUC) Beyond this, it must be noted that the underlying mechanisms in kidney disease are not completely understood in patients with severe kidney impairments, and thus they require a significantly lower dosage [90,91]. In other words, even those drugs that are eliminated by nonrenal metabolism and transport could have unintended effects if they are prescribed without taking into account the necessary adjustments for reduced renal function. Previously undertaken studies in animals suggest that in the presence of kidney impairment, metabolic enzymes and transporters in the liver and intestine are downregulated or upregulated, and thus extend their influence to nonrenal clearance drug responses.

## 8. Strategies Aimed at the Removal of UTs

New membrane-based strategies have been proposed for the improvement of dialytic removal of PBUTs, but to date, longer dialysis sessions and hemodiafiltration have only led to mild improvements. The use of larger dialyzers in combination with higher dialysate flow rates provided some efficiency in indoxyl sulfate (IS) clearance, but with reduced impact on the patients’ quality of life. Fractionated plasma separation and adsorption (FPSA) proved to be twice as efficient as traditional HD in eliminating IS and p-cresol sulfate (PCS), although there was a risk of occlusive thrombosis.

A promising technique for the removal of PBUTs is binding competition using pharmaceutical drugs as displacers [92,93]. This technique holds out great hopes for future haemodialysis. Tao et al. [93] tested, via rapid equilibrium dialysis with HAS, a few pharmaceutical drugs and evaluated their competition with albumin binding sites (Figure 10). The results showed that ibuprofen (IBU) possesses the highest binding affinity among the tested displacers. The dosage of IBU followed the usual prescription concentrations, and a higher removal for higher doses was observed. The work also showed ex vivo assays in human blood when ibuprofen was infused upstream in the dialyzer into the blood compartment. A proof of concept of enhanced PBUTs removal, particularly a threefold increase in IS removal during HD by infusion of binding competitors, IBU and FUR, upstream in the dialyzer, was demonstrated (Figure 11). Moreover, non-protein-bounded toxins, represented by urea, are not affected by the infusion.

Another clinical study [94] designed a proof of concept to explore the utility of IBU competitive binding in patients on HD maintenance. The HD treatment included three phases: pre-infusion (1–20 min), an ibuprofen infusion phase (21–40 min), and post-infusion (41–240 min). A dose of 800 mg of IBU was infused at a constant rate into the arterial bloodline. Figure 12 shows a schematic concept of the dialyzer referred to in this study, where the UTs enter bounded to albumin in blood inflow and eight contacts with the displacer (IBF). It was found that IBF displaced the PBUTs, allowing their removal by the dialysis membrane. The enhanced removal of IS and PCS was effectively observed in all studied patients. The results showed a twofold and fourfold increase in the concentration of IS and PCS, respectively, in the dialysate outflow during the infusion phase, reflecting a good efficacy of the competitive binding strategy.

IBU only competes with one of the binding sites in HSA. This could be overcome by the use of more than one drug during infusion, a solution that was analyzed by infusing a mixture of salvianolic acids in Sprague-Dawley rats for 4 h, where the first 2 h served as control. The IS and PCS removal improved by 135.6% and 272%, respectively [95]. This study highlights an important point regarding the use of a binding competitor cocktail that targets multiple binding sites on BSA, which leads to higher PBUTs removal rates during dialysis [92].

In summary, there have been several reports on the clinical approach being tested by physicians to increase PBUT clearances, which involve the use of binding competitors to displace UTs from HSA. All of the studies showed that PBUT clearance is increased when large doses of pharmaceutical drugs, such as IBU, are injected into the bloodstream of patients during HD [96]. Despite promising results, systemic IBU delivery to patients with CKD is not recommended, as it has been associated with deterioration of residual kidney function, nephrotoxicity, hypertension, and gastrointestinal bleeding [97]. It is evident that this approach poses many risks to patients and cannot be adopted in thrice-weekly HD sessions.

Faria et al. [16,98,99,100,101,102] believe the use of drugs can enhance PBUT clearances in a safe and effective way if they are strategically located at the surface of the HD membrane. This way, when blood enters the hemodialyzer, binding competitors displace the UTs locally, which are then easily and quickly filtered across the HD membrane. The main goal of this research group is to enhance the PBUT clearance of monophasic hybrid cellulose acetate-based membranes by surface functionalization with binding competitors.

Another strategy was purposed by Stamatialis’s group [103,104,105] based on a mixed matrix membranes (MMM) approach. Here, MMM, made of synthetic materials, are dual-layered hollow fiber membranes consisting of an inner layer composed of a polyethersulfone (PES) and polyvinylpyrrolidone (PVP) blend and an outer layer composed of activated carbon (AC) microparticles embedded in PES/PVP. The inner layer is responsible for transport selectivity, which should be thin enough to minimize mass transfer resistance and to protect the patient’s blood from contacting the sorbent particles. The activated carbon component in the outer layer promotes efficient adsorption and the removal of UTs, maintaining a high toxin concentration gradient between the blood and the dialysate solution, resulting in a higher removal of the UTs. Additionally, the efficiency of the toxin adsorption can be adapted by applying different types of sorbents. The MMMs combine the benefits of diffusion and convection, provided by the membrane structure, and absorption achieved by the AC microparticles dispersed through the membrane [103]. In addition to providing high UTs removal for surrogate markers of uremic toxicity such as creatinine [104] and indoxyl sulfate [103], high flux dual-layered MMM (which ensure the effect of convective solute transport during HD) with low albumin leakage have been developed, with dimensions and selectivity similar to commercial membranes [105].

## 9. Conclusions

There is compelling and alarming evidence that UTs pose a real threat to cardiovascular health. The links between heart and kidney health are strong and are no doubt fortified by as-yet unknown molecules and mechanisms. The study of UTs, and their molecular and cellular effects may establish new biomarkers and paint new therapeutic targets in sufferers of CKD. Undoubtedly, UTs remain a critical challenge.

One recommended strategy to remove PBUTs is to apply pharmaceutical drugs as displacers, based on competitive binding. One strategy is the removal of PBUTs through a mixed matrix membranes (MMM) approach. To achieve a higher removal of UTs, some significant factors, including flux dual-layered MMM, albumin leakage with dimensions, and selectivity similar to commercial membranes, should be optimized. A recently undertaken response employs drugs that are carefully localized on the surface of new HD membranes. The door is still open for novel materials and scientific/pharmaceutical developments.

## Figures and Tables

**Figure 1 membranes-12-00261-f001:**
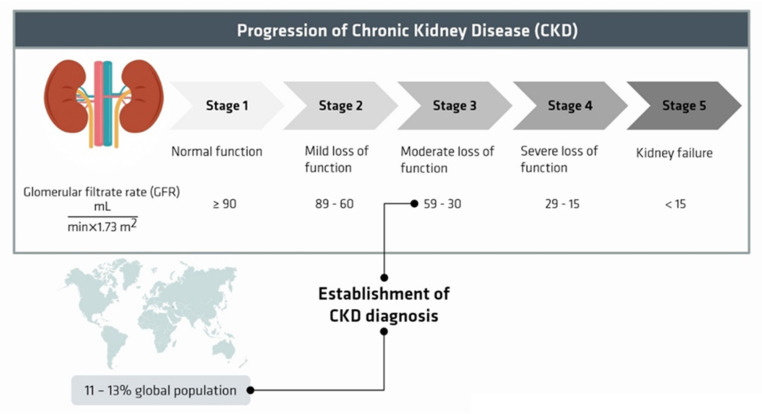
Degrees of decreased kidney function, adapted from [1].

**Figure 2 membranes-12-00261-f002:**
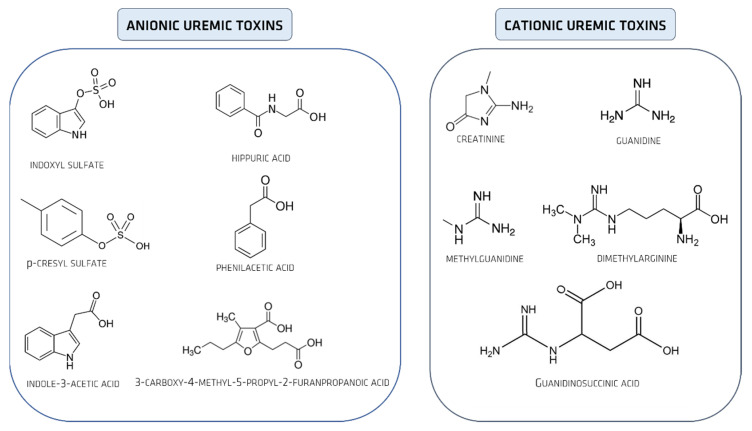
Chemical structures of anionic and cationic UTs (adapted from [6]).

**Figure 3 membranes-12-00261-f003:**
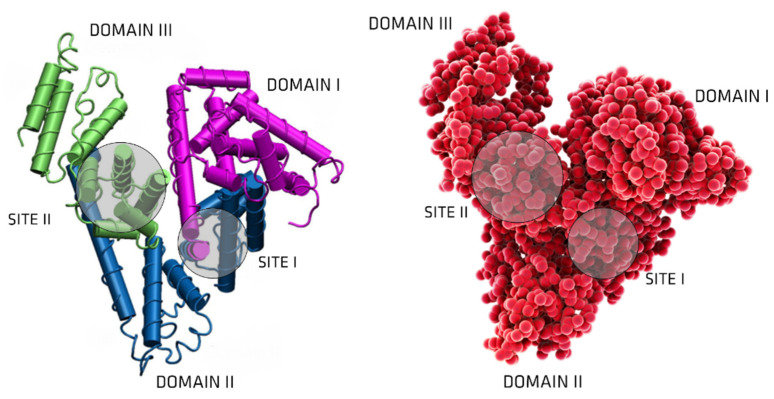
HAS structure. The primary binding sites for anionic, neutral, and cationic ligands are commonly referred to as Sudlow’s sites I and II. In general, bulky heterocyclic anion ligands generally bind to Sudlow’s site I (or the warfarin site), whereas Sudlow’s site II (or the indole-benzodiazepine site) ligands are aromatic and can be either neutral or bear a negative charge located peripherally on the molecule (adapted from [27]).

**Figure 4 membranes-12-00261-f004:**
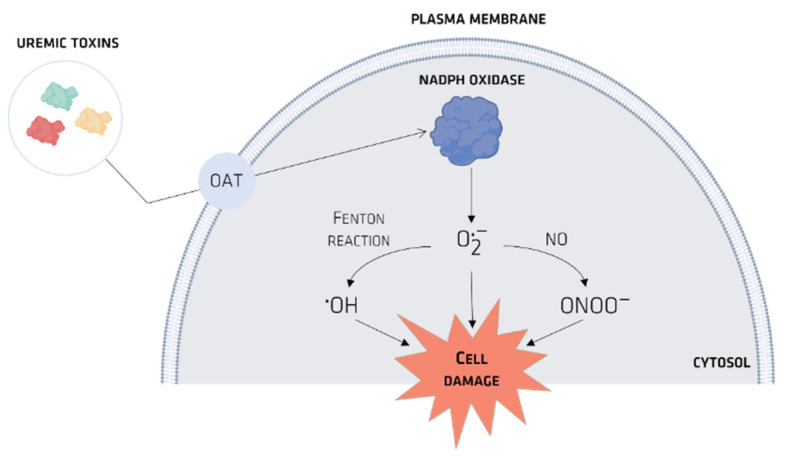
UTs-induced ROS production.

**Figure 5 membranes-12-00261-f005:**
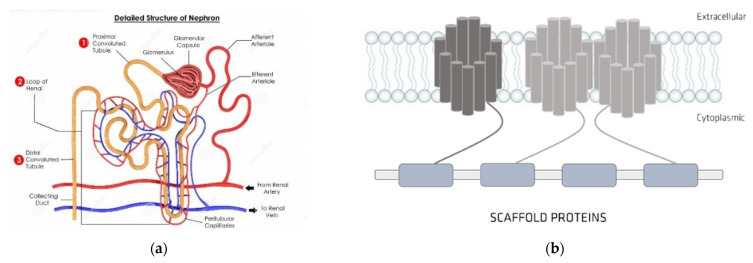
(**a**) nephron structure; (**b**) organic anionic transporter’s structure.

**Figure 6 membranes-12-00261-f006:**
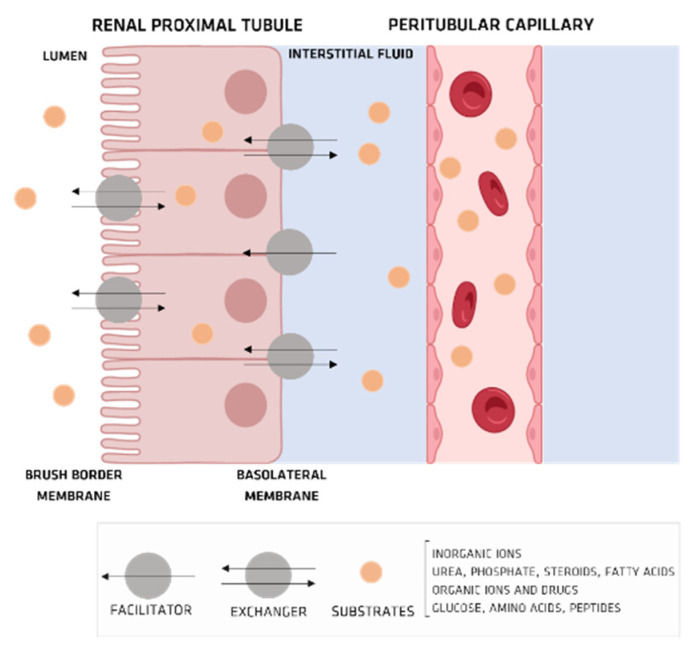
The roles of transporters in the human kidney.

**Figure 7 membranes-12-00261-f007:**
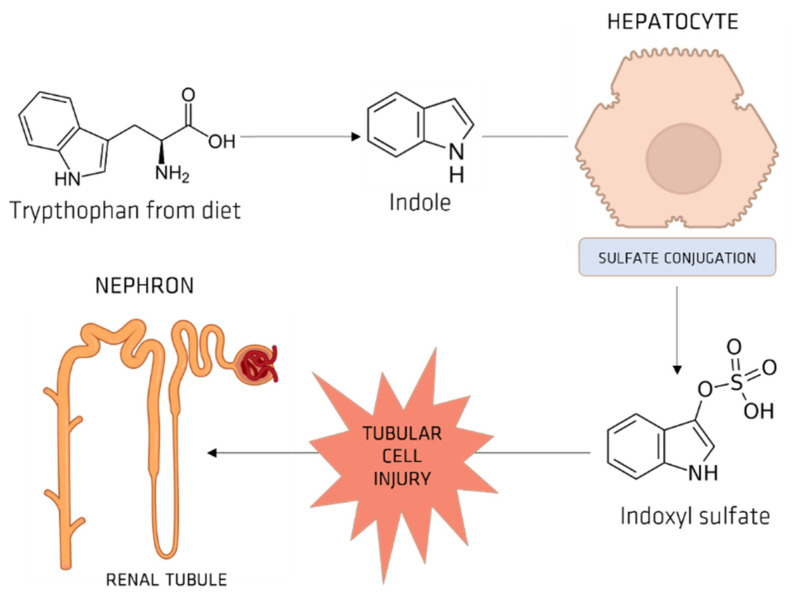
IS promotes the progression of CKD.

**Figure 8 membranes-12-00261-f008:**
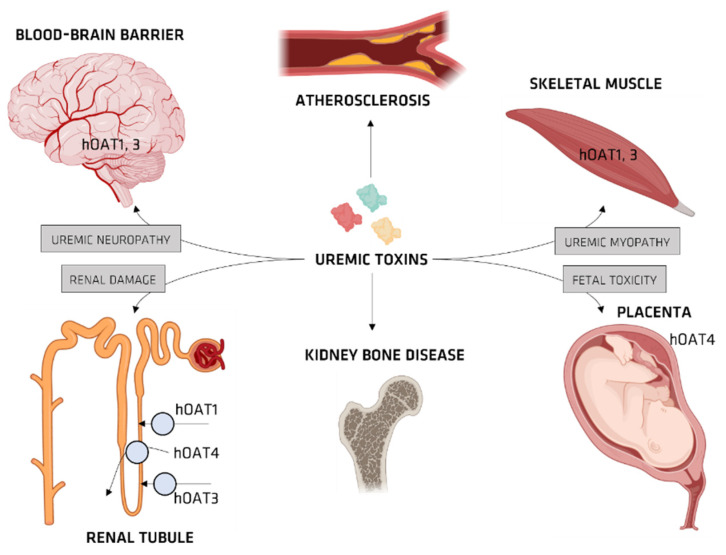
Pathological role of organic anion transporters (OATs) in the progression of uremia, atherosclerosis, and kidney bone disease.

**Figure 9 membranes-12-00261-f009:**
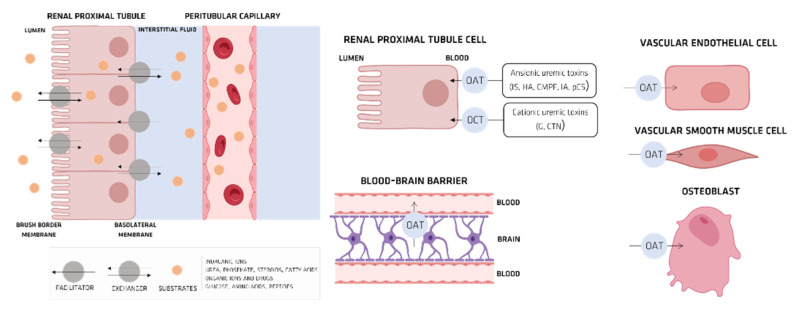
OAT- or OCT-mediated cellular transport of anionic and cationic uremic toxins (adapted from [6]).

**Figure 10 membranes-12-00261-f010:**
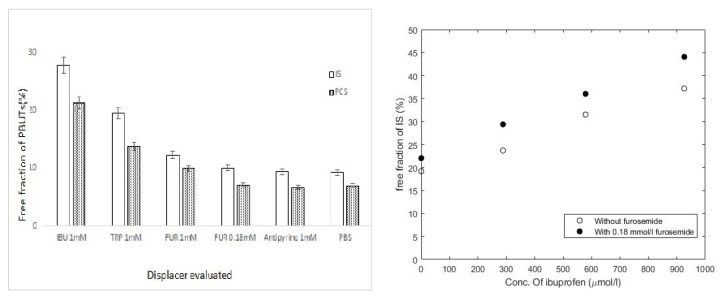
IS and PCS displacement in uremic plasma by furosemide (FUR), tryptophan (TRP), and IBU (**left**), and the IBU and FUR dosage effect in IS displacement (**right**), both determined in static RED assays [93].

**Figure 11 membranes-12-00261-f011:**
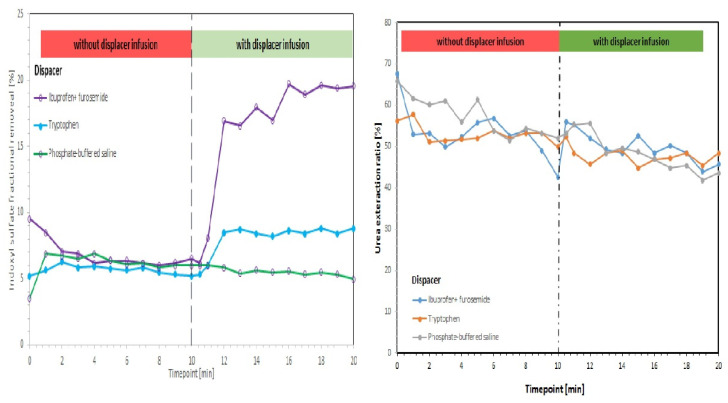
Human whole blood experiments with a competitive binding strategy showing IS removal (**left**) and urea removal (**right**) (adapted from [93]).

**Figure 12 membranes-12-00261-f012:**
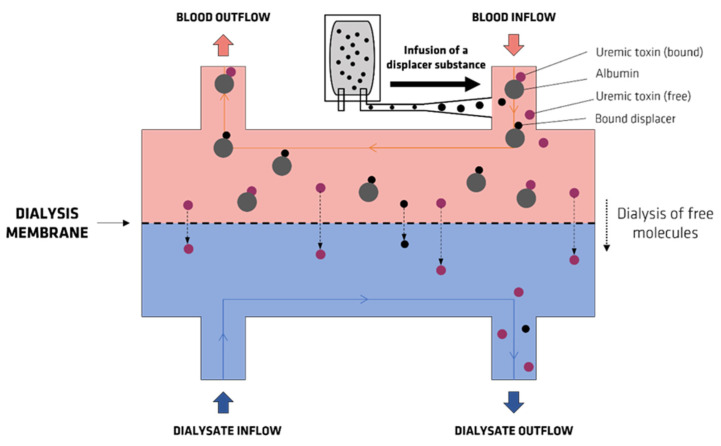
Schematic concept of the dialyzer used in a previous work by Madero et al. (adapted from [94]).

**Table 1 membranes-12-00261-t001:** Retention solutes first group: low molecular weight (<500 D) and non-known protein binding.

	MW (Da)	Average C_U_ (mg/L)	Average C_N_ (mg/L)	Maximum C_U_ (C_MAX_) (mg/L)	Group	References
ADMA mg/L	202	1.6 ± 1.2/10	0.2 ± 0.06/6	7.3	Guanidines	[7,8,9]
Creatine mg/L	131	134.0 ± 30.3/29	9.7 ± 3.3/24	235.8	Guanidines	[7,10]
Creatinine mg/L	113	136.0 ± 46.0/19746	<12.0/23	240.0	Guanidines	[7,11,12]
Guanidine µg/L	59	172.9 ± 83.8/13	<11.8/16	800.0	Guanidines	[7,13,14]
SDMA µg/L	202	640.3 ± 212.1/38	76.1 ± 21.0/66	1232.2	Guanidines	[7,15]
Urea g/L	60	2.3 ± 1.1/16	<0.4/23	4.6		[7,12]

Abbreviations are C_N_, normal concentration; C_U_, mean/median uremic concentration; C_MAX_, maximal uremic concentration; MW, molecular weight. Normal values are reported as means ± SD; C_MAX_ values were calculated as mean + 2 SD based on C_U_). ADMA = asymmetrical dimethylarginine; SDMA = S symmetrical dimethylarginine.

**Table 2 membranes-12-00261-t002:** Retention solutes second group: low molecular weight (<500 D) with known protein binding.

	MW (Da)	Average C_U_ (mg/L)	Average C_N_ (mg/L)	Maximum C_U_ (C_MAX_) (mg/L)	Group	References
CMPF mg/L	240	61.0 ± 16.5/15	7.7 ± 3.3/7	94.0		[7,16,17]
HA mg/L	179	247.0 ± 112.0/7	<5.0	471.0	Hippurates	[7,16,18]
IA µg/L	175	875.0 ± 560.0/42	17.5 ± 17.5/7	9076.9	Indoles	[7,16,19,20]
IS mg/L	251	53.0 ± 91.5/20	0.6 ± 5.4/40	236.0	Indoles	[7,17]
PCS mg/L	108	20.1 ± 10.3/20	0.6 ± 1.0/12	40.7	Phenols	[7,16,21]

Abbreviations are C_N_, normal concentration; C_U_, mean/median uremic concentration; C_MAX_, maximal uremic concentration; MW, molecular weight. Normal values are reported as means ± SD; C_MAX_ values were calculated as mean + 2 SD based on C_U_). CMPF = 3-carboxy-4-methyl-5-propyl-2-furanpropionic acid; HA = hippuric acid; IAA = indoleacetic acid; IS = indoxyl sulfate.

**Table 3 membranes-12-00261-t003:** Retention solutes third group: middle molecular weight.

	MW (Da)	Average C_U_ (mg/L)	Average C_N_ (mg/L)	Maximum C_U_ (C_MAX_) (mg/L)	Group	References
Leptin µg/L	16,000	72.0 ± 60.6/8	8.4 ± 6.7/56	490.0	Peptides	[7]
Neuropeptide Y ng/L	4272	64.9 ± 25.5/19	<80.0	115.9	Peptides	[7,22]
Parathyroid hormone µg/L	9225	1.2 ± 0.6/10	<0.06	2.4	Peptides	[7,23]
Retinol-binding protein mg/L	21,200	192.0 ± 78.0/112	<80.0	369.2	Peptides	[7,24]

Abbreviations are C_N_, normal concentration; C_U_, mean/median uremic concentration; C_MAX_, maximal uremic concentration; MW, molecular weight. Normal values are reported as means ± SD; C_MAX_ values were calculated as mean + 2 SD based on C_U_).

**Table 4 membranes-12-00261-t004:** Pharmacokinetics and redox properties of anionic UTs.

Anionic UTS	Intracellular UPTAKE by	Albumin Binding	Ease Removal of HD	ROS Production	Effect on Nonrenal Clearance
IS	OAT1, OAT3	High affinity (site II)	Partially	Yes	Inhibition of rCYP3A, hCYP2C9, hCYP1A, hCYP3A, AHR ligand
CMPF	OAT1, OAT3	High affinity (site I)	Difficult	Yes?	Inhibition of rOatp2, rOatp1a4
PCS	OAT1, OAT3	High affinity (site II)	Partially	Yes	-
HA	OAT1, OAT3	High affinity (site II)	Partially	unknown	-
IA	OAT1, OAT3	High affinity (site II)	Partially	Yes	-

## Data Availability

Not applicable.

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
