# Peer review of "Interaction of Human Serum Albumin with Uremic Toxins: The Need of New Strategies Aiming at Uremic Toxins Removal"

_membranes, 2022, doi:10.3390/membranes12030261_

Round 1

Reviewer 1 Report

In this work authors discuss the interactions of protein-bound uremic toxins with the transport protein - human serum albumin (HSA). The work is interesting and earns probably high interest justifying publication in the Membranes. The references are upto date and comprehensive, the structure of the manuscript is appropriate, the data collected by the techniques applied support the conclusions. I suggest publication as it is or subject to the following minor remarks:

From line 407 authors discussed the competitive binding of different UTs to HSA. My opinion is that statements here are the main messages of this work. The stability (binding) constants listed in lines 413 and 414 reflect weak interactions between the toxins and the HSA (in agreement with the related theories). However, in the transfer process the key is the fetching and the releasing of the drug by the albumins, which however several times associated with the change of the molecular environment: i.e. interactions whose are strong in hydrophobic environment can be weak in the hydrophilic environment and sometimes the binding sites also exchanges since the molecular environment differently affect the binding property in a given sites. I feel importance to insert few sentences about how the molecular environment can affect the statements formed about the competitive binding of the drugs onto the surface of albumins.       

Author Response

The authors do agree with the referee #1 comment, and accordingly added few sentences (supported by new references) to the manuscript.

Reviewer 2 Report

The main aim of this review to summarize the removal methods of  uremic toxins. The novelty of this review is to collect and critically evaluate the new strategies.

The authors cited and reviewed 110 references, prepared 12 figures, 4 tables.

The review is very well-written, but review design figure is needed to support the visibility and understanding of the content.

Author Response

Authors

The authors do agree with the referee #2 comment, and accordingly added the following Figure (as Graphical Abstract) to the manuscript.
